# A bibliometric analysis of the gender gap in the authorship of leading medical journals

Oscar Brück [1✉]

## Abstract

**Background** Monitoring gender representation is critical to achieve diversity and equity in academia. One way to evaluate gender representation in academia is to examine the authorship of research publications. This study sought to determine the gender of first and senior authors of articles in leading medical journals and assess trends in the gender gap over time.

**Methods** We gather bibliometric data on original research articles ($n = 10,558$) published in 2010–2019 in five leading medical journals to audit publication and citation frequency by gender. We explored their association with scientific fields, geographical regions, journals, and collaboration scope.

**Results** We show that there are fewer women as senior (24.8%) than leading authors (34.5%, $p < 0.001$). The proportion of women varied by country with 9.1% last authors from Austria, 0.9% from Japan, and 0.0% from South Korea. The gender gap decreased longitudinally and faster for the last ($-24.0$ articles/year, $p < 0.001$) than first authors ($-14.5$ articles/year, $p = 0.024$) with pronounced country-specific variability. We also demonstrate that usage of research keywords varied by gender, partly accounting for the difference in citation counts.

**Conclusions** In summary, gender representation has increased, although with country-specific variability. The study frame can be easily applied to any journal and time period to monitor changes in gender representation in science.

## Plain language summary

The publishing of medical research papers has traditionally been dominated by men. To better understand whether gender diversity in the authorship of research papers has changed recently, we analyzed over 10,000 articles published between 2010 and 2019 in five top medical journals. Usually, the first author is recognized as the leading contributor, whereas the last author supervises the study. We found that there were fewer women in senior positions compared to first author positions. The percentage of women as authors varied across countries. Over time, the gender gap decreased, but at different rates depending on the author's position and country. Keywords selected by researchers to describe their work varied between genders. Our findings show progress in gender representation, but with country-specific differences. This study can be used as a model to track gender representation in other journals and time periods.

[1] Hematoscope Lab, Comprehensive Cancer Center & Center of Diagnostics, Helsinki University Hospital, Helsinki, Finland & Department of Oncology, University of Helsinki, Helsinki, Finland. ✉email: oscar.bruck@hus.fi

Gender equity in medical research refers to the equal representation, recognition, and valuation of all researchers regardless of their gender. Women and other underrepresented groups continue to face barriers in diverse research fields. The proportion of graduating female medical students is equal to that of men[1]. Yet, publications are more frequently authored[2,3], peer-reviewed[4], and editorially evaluated by men[5,6]. Based on a report by the Association of American Medical Colleges, the proportion of women faculty had increased to 41% in 2019, while their proportion as department chairs remained at 18%[7].

Gender underrepresentation in scientific publishing has been documented earlier. Women tend to obtain university-level degrees more commonly than men in OECD countries (47% vs. 32%)[8]. However, female doctoral students submit and publish less than their male colleagues[2,3]. The difference is largest in natural/biological sciences and engineering and has been hypothesized to result in unevenly distributed resources and support[3]. Gender disparities in publications have been shown to correlate with future academic rank signifying sustained impact on professional careers[9], salary and job satisfaction[10]. Instead, pay transparency has been recently shown to significantly reduce inequity[11].

Abrogating the gender gap in all levels of academia are official priorities of the National Institutes of Health[12] and the European Commission[13]. Promoting gender equity can diversify science and be a critical step in achieving both academic and technological breakthroughs. In addition, it can help to reduce bias and discrimination in academic careers by ensuring that all researchers are treated fairly and given equal opportunities to succeed.

The representation of women in leading and senior publication authorship positions remains disproportionately low[14,15]. Medical journals with high journal impact factors (JIF) have not been comprehensively examined, although these share demanding editorial and peer-review assessment, and priority for groundbreaking science. We recently discovered previously undocumented Anglocentric bias in leading medical journals based on publication counts and their citation frequency[16]. Here, we reasoned that gender underrepresentation could be objectively explored with a similar retrospective approach. We demonstrate increased gender representation in journal authorships. However, gender representation is tightly associated with both research fields, geographical regions, and scientific journals. The study frame can be reapplied to other journals and time periods to track changes in gender representation in science.

## Methods

**Data collection**. For this study, we collected data from medical journals that were ranked highest in the Journal Citation Reports 2022, published primarily original articles and within overlapping scientific scopes (Fig. 1a). *The New England Journal of Medicine* (NEJM), *Nature Medicine* (NatMed), *Journal of the American Medical Association* (JAMA), *The BMJ*, and *The Lancet* met these criteria. We included all original articles published from 2010 to 2019, totaling to 10,558 articles. We excluded more recent publications to avoid bias related to the COVID-19 pandemic and to ensure equal opportunities for accumulating citations. We queried for articles in the Web of Science database by Clarivate Plc with the terms "(((SO = (NATURE MEDICINE OR LANCET OR NEW ENGLAND JOURNAL OF MEDICINE OR JAMA JOURNAL OF THE AMERICAN MEDICAL ASSOCIATION OR BMJ BRITISH MEDICAL JOURNAL)) AND DT = (Article)) AND PY = (2010-2019))". This allowed us to download metadata for each article, including the names of the authors, the address of the corresponding author, and the total number of citations.

**Author name prediction**. To determine the number of authors, we summed the frequency of the semicolon ";" delimiter between author names and then added 1. All author names were available in Latin alphabet. The bibliometric data in the Web of Science does not specify possible shared first or last authorships. Therefore, our analysis interpreted first and second author names based on their sequential authorship order. For author names, only the initials of the forename were available for 538 first (5.1%), 529 s (5.0%), 780 s last (7.4%) and 572 last authors (5.4%). In total, we identified 7558 unique forenames and predicted the gender for 7113 (94.1%) forenames with the genderizeR library based on the genderize.io database. While the database employs 'male' and 'female' for sex, here, the term 'gender' is used, acknowledging the challenge of accurately determining gender or sex solely through genderization. However, the notion of predicted gender ("man" or "woman") refers neither to the sex of the authors nor the gender that the author self-identifies as. To avoid bias related to names from certain countries, we applied a low (>50%) probability threshold.

**Geolocation**. To determine the geographic location (latitude and longitude coordinates) of the primary institutes where the research was conducted, we applied the ggmap library and Google's Geocoding API to the corresponding author's address. If we were unable to successfully match the address, we geolocated the text-mined city and country of the address. The combined approach resulted in the successful geolocation of 10,730 out of 10,732 total unique addresses (100.0%).

**Keyword analysis**. In the keyword analysis, we identified 21,820 unique keywords. As most of these were rarely used, we included only keywords employed in at least 20 articles (n = 583, 2.7%).

**Statistical analysis**. We measured the impact of an article by its average citation count per year. We used the Wilcoxon rank-sum test (unpaired, two-tailed) to compare two continuous variables and the Kruskal-Wallis test to compare three or more continuous variables. For categorical variables, we used the $\chi 2$ test. We adjusted p values using the Benjamini–Hochberg correction. To compare two linear regression slopes, we tested the significance of the interaction term using T test. All statistical analyses and visualizations were conducted using R 3.5.1. using base, tidyverse, fastDummies, maps, reshape2, ggmap, data.table, countrycode, ggpubr, ggrepel, rstatix, ggdendro and dendextend libraries.

**Reporting summary**. Further information on research design is available in the Nature Portfolio Reporting Summary linked to this article.

## Results

**The gender gap in productivity is declining but not the gap in citation count**. First, we interrogated how authoring patterns would differ between women and men in original articles published in 2010–2019 in five leading medical journals (Fig. 1a). Publications with a man as first author were two times and with a man as last author three times as common compared to publications authored by a woman as first and last author, respectively (Table 1). In addition, publications with women as first authors were 46.8% more likely to have a woman as last author compared to publications with a man as first author in line with a previous report ($\chi 2$ p < 0.001, Fig. 1b)[17].

Temporally, the publication count has gradually converged between women and men (Fig. 1c). We fitted a linear regression for publication count using publication year and gender as covariate. The interaction term of the regression model was significant (coefficient −14.5, p = 0.024) indicating that the yearly

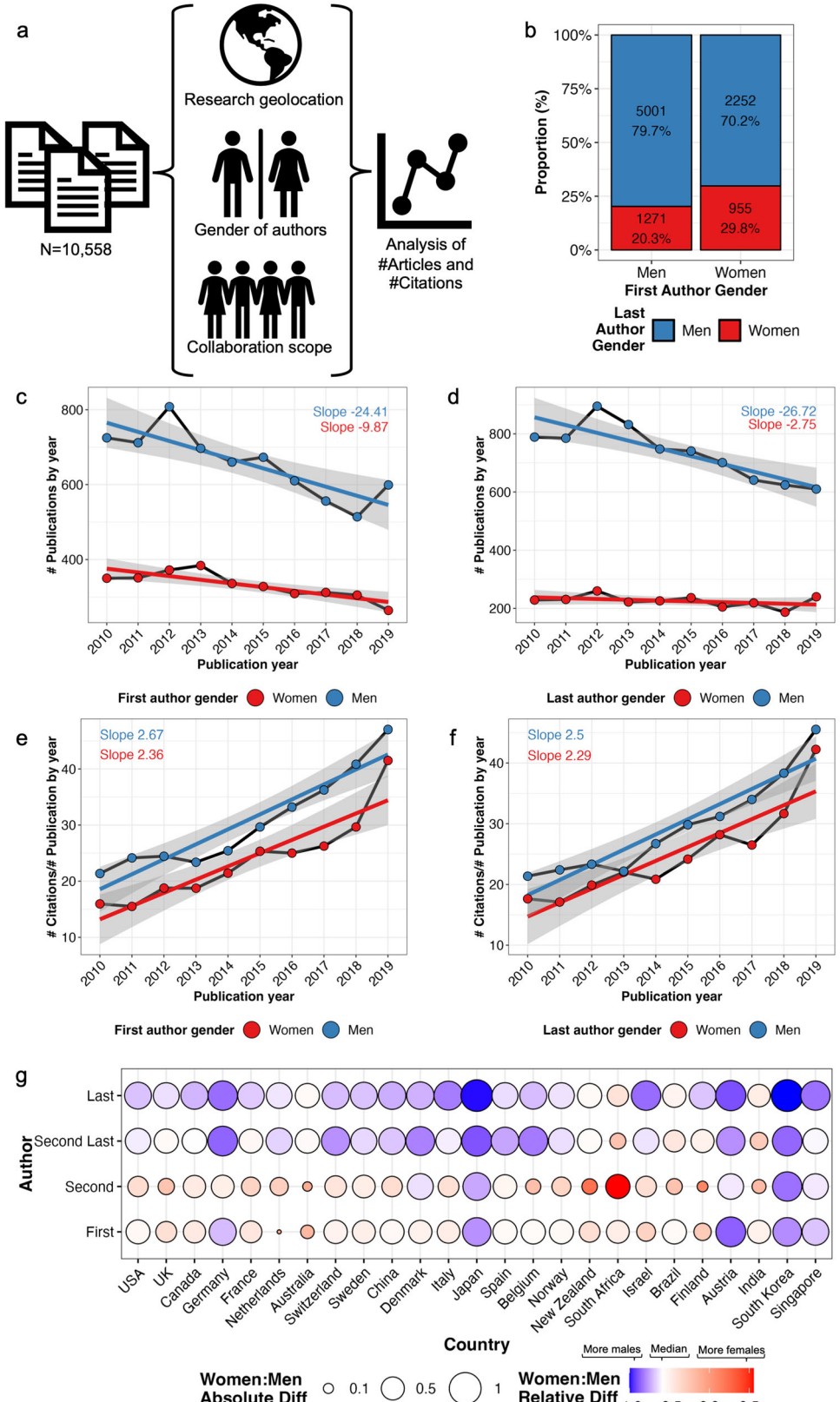

**Fig. 1 Gender gap in medical publishing. a** Study design. **b** Bar plot on the association between first and last author gender. **c**, **d** Line plot and fitted linear regression for the number of publications and (**e**, **f**) number of yearly-averaged citations per publication by their publishing year and author gender. The shaded area represents the 95% confidence interval of the regression model. **g** Association of the first, second, second last and last author gender distribution by the affiliation nationality of the corresponding author. The balloon color reflects the gender bias. To emphasize differences white color defines the median, i.e. men than women authors (red = more women, orange = balanced, blue = more men). The balloon size reflects the absolute deviation from a balanced gender distribution.

**Table 1 Association of gender authorship with publication metrics. Median and 25–75% interquartile ranges are reported.**

| Authorship | Variable | Women | Men | P-value |
|---|---|---|---|---|
| First | # Publications | 3311 | 6554 | |
| First | # Citations | 22.0 [10.9–46.4] | 29.0 [13.6–60.9] | *** |
| First | # Authors | 10.0 [6.0–17.0] | 11.0 [6.0–18.0] | *** |
| Last | # Publications | 2256 | 7366 | |
| Last | # Citations | 23.5 [11.0–52.4] | 28.2 [13.4–58.3] | *** |
| Last | # Authors | 10.0 [5.0–17.0] | 11.0 [6.0–18.0] | *** |

***p < 0.001.

decline of first authorships for men was higher in comparison to women. The decline was even more distinct for last authors (coefficient −24.0, p < 0.001 for the interaction term; Fig. 1d). Collectively, the gender gap in top medical research has declined with an average 14.5 publications/year for the first authors and 24 publications/year for senior authors.

Articles authored by women gathered also fewer citations (Table 1). The median citation frequency by publication increased in line with rising medical journal impact factors (Fig. 1e, f). However, the ascent was equal between women and men for both first (p = 0.53 of the interaction term between publication year and gender; Fig. 1e) and last authors (p = 0.67; Fig. 1f) implying no convergence in accumulated citations.

**The gender gap in productivity varies by country**. The proportion of women authoring in leading medical journals varied by the country where the research had been conducted (Fig. 1g). Women as authors were least common in publications originating from South Korea, Japan, Singapore, Germany, and Austria, whereas they were most frequent from South Africa and India. The gender gap was more pronounced for second last and last authors with 24.8% women in the median across countries compared to first and second authors with 34.5% women (p < 0.001). Gender underrepresentation was highest in South Korea with 0.0% (0/28), Japan with 0.9% (1/106) and Austria with 9.1% (3/33) women as last authors.

Next, we examined longitudinal national trends in gender representation. The gender-associated difference in publication number decreased rapidly in UK, Canada, and Belgium both for the first and last authors (Supplementary Fig. 1a, b). Yet, the gender gap remained evident at the end of the follow-up in many countries, such as in the USA, UK, and China, and particularly at the last author level. In opposite to the general trend, the gender gap rose for last authors of publications from Israel and both for the first and last authors from China. Collectively, these findings imply that the longitudinal trend could guide in customizing national measures to mitigate gender underrepresentation.

**Collaboration scope varies by gender and is associated with publication and citation count**. As the number of co-authors (R 0.46, p < 0.001) correlated with citation count, we interrogated next the significance of collaboration scope over gender-related differences in citation count. On average, publications with men as first or last author included one co-author more than those with women as first or last author (Table 1). However, when applying a linear regression model using gender, author count and their interaction term as covariates, these were all observed to be independent but weak predictors of future citation count (Supplementary Table 1).

When examining the temporal evolution of authorship patterns, we observed a stable median increase of 9 additional authors per article during the 10-year follow-up (Supplementary Fig. 2a).

Publications with more than 11 authors doubled and these accumulated faster citations per article (2.99 citations/article/year vs. 1.19 citations/article/year; Supplementary Fig. 2b). The inclination was also 55.3% steeper for men as first author and 94.1% for men as last authors during the 10-year follow-up compared to corresponding proportions with women (Supplementary Fig. 2c). Instead, the slope between first and last authors did not differ when comparing separately women and men (Supplementary Fig. 2c).

To study national differences in collaboration scope, we compared the number of authors for publications by gender and by country (Fig. 2a). Publications from USA, UK, Denmark, and Sweden shared the least authors (Fig. 2a). On the opposite, publications from France, Germany, China, and South Korea were associated with more numerous authors/article (Fig. 2a). Together, the findings emphasize that the geographical origin of the research has a more pronounced association with collaboration scope than gender.

**Thematic and journal disparities between women and men**. Previous reports have suggested differences in funding, mentorship training, and household and caregiving responsibilities between women and men to account for the gender gap in medical publishing[2,18]. We hypothesized whether the areas of research would differ by gender. We observed a clear correlation between keyword-associated citation frequency and the proportion of men as first (R 0.40, p < 0.001) and last authors (R 0.40, p < 0.001; Fig. 2b). Keywords associated with the highest publication citations were related to phase II-III trials, oncology, immunotherapy, chemotherapy, and antibody-based therapy, and were enriched with publications authored by men, especially at the senior author level. On the contrary, the 20 keywords associated with a least citations were predominantly associated with publications authored by women, notably in the context of first authorship (Fig. 2b). The keywords covered healthcare-related themes such as patient involvement, insurance, quality-of-care, and access.

Beside gender-associated distinctions in research subfields, we sought to understand whether publications sharing the same keyword would differ by their accumulated citation count. Out of 583 keywords, we included 579 for comparing first author and 582 for the last author gender comparison as these keywords were used by both genders. For the first authors, 78/579 compared to 5/579 keywords resulted in higher citation count (adjusted p < 0.05) when employed by men vs. women, respectively. For last authors, the corresponding proportion was 32/582 for men and 8/582 for women (adjusted p < 0.05). Collectively, while the finding is relevant in distinct fields, no association between gender and citation count were observed for 85% of keywords when comparing first authors by gender and 93% of keywords when comparing last authors by gender.

To conclude, we investigated the proportion of publications authored by women in distinct journals. The absolute difference in the proportion of first authors between the five journals was 14.8% for first and 10.2% for last authors (Fig. 2c). Women as first and second authors were least frequent in articles published in *NEJM* and *Lancet*, whereas second last and last authors were least common in *NatMed*. According to this analysis, *BMJ* and *JAMA* were the most representative journals considering all four authorship positions.

## Discussion
Available bibliometric data can reveal novel information on the equity and diversity of scientific research. Here, we presented publication disparity in five leading medical journals between 2010 and 2019.

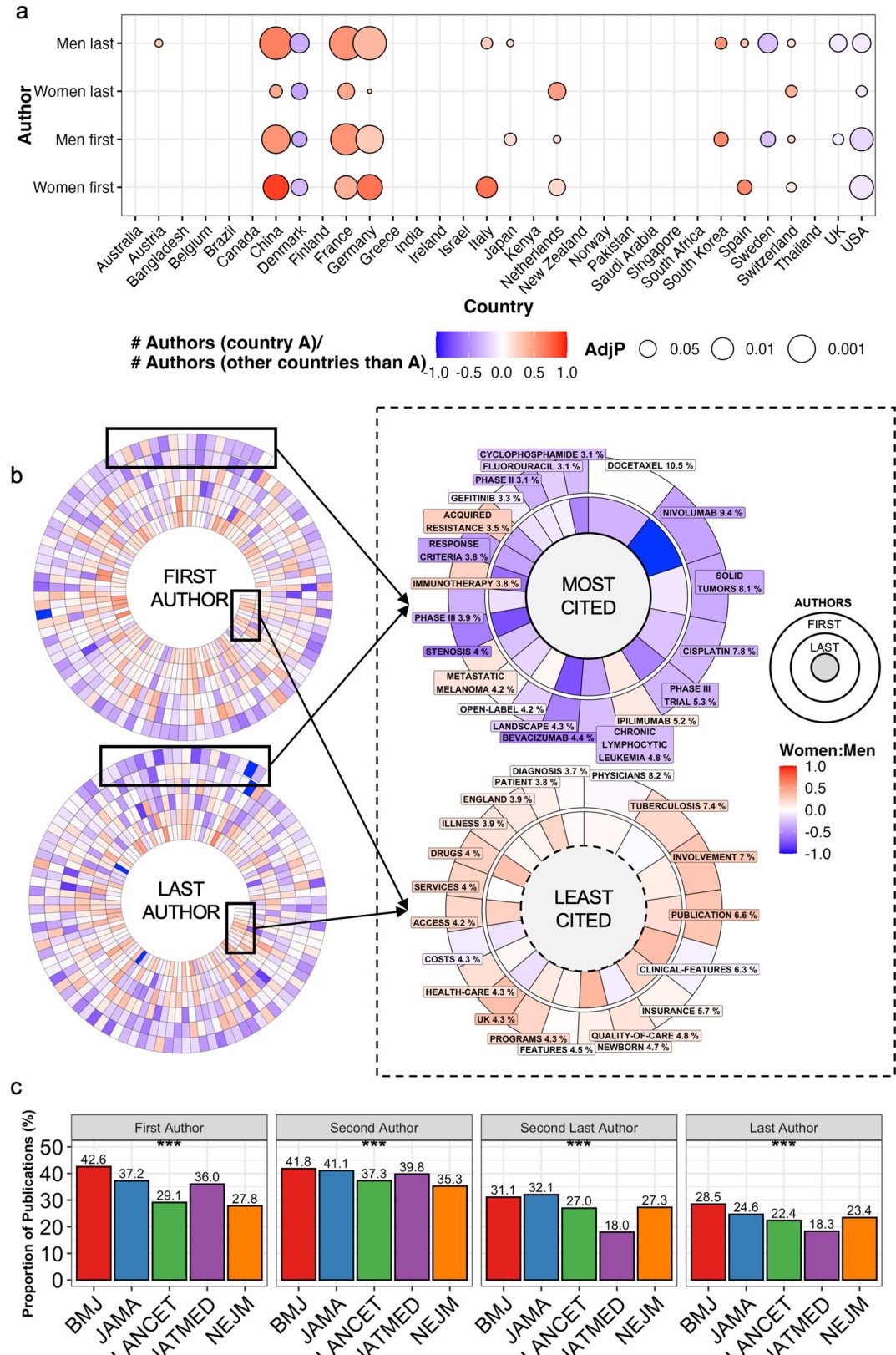

**Fig. 2 Gender gap by collaboration scope, subfields and journals. a** Association of the first and last author gender distribution by the affiliation nationality of the corresponding author. The balloon color reflects the number of authors by country (column) compared to all other countries and the balloon size the adjusted *p* value of that comparison. **b** Left-side spiral plots illustrate the keywords occurring ≥ 20 times and arranged by their citation impact starting from the outside layer (most cited) towards the inner core (least cited). Right-side plots show the proportion (size of the bar) of the most and least cited keywords. The color of the bars in all spiral plots illustrate the gender balance. **c**. Bar plot illustrating the proportion of publications in medical journals originating from different countries.

Our findings in leading medical journals are in line with previous observations. For the first time to our knowledge, we report that the inequity is country-specific. While the distinction is visible across first, second, second last and last author positions, the gender gap is most pronounced at the senior author level. In particular, publications with a corresponding author in Germany, Austria, Japan, South Korea, and Singapore had the fewest women as authors. The finding implicate that the bias likely arises from national and cultural factors rather than editorial or peer-review processes, which had previously indicated mixed results[19,20]. Moreover, the gender gap in first and last authorships has steadily declined during the last decade in most countries, but the progress has been less evident or even contrary in some countries such as China and Israel. The gender disparity has decreased more rapidly in last authorships possibly reflecting a response to the more pronounced gender underrepresentation compared to first authors or dynamics in generation transition ("demographic inertia"). This phenomenon could be due to a gradual increase of women in senior academic positions, while being rare in the past.

Gender underrepresentation was observable in all top medical journals. However, women as first and second authors were least frequent in articles published in *NEJM* or *Lancet*, whereas second last and last authors were least common in *NatMed*. The findings are in line with a previous study examining gender representation as first authors using 4 out of 5 similar journals during 1994–2014[21]. The trend was replicated also in an article examining author disparity in leading medical journals during the COVID-19 pandemic[18]. In that study, no difference was found when examining first and last author gender of COVID vs. non-COVID-related research[18]. Confirming our longitudinal findings, the proportion of women as first (36.2% vs. 33.6%) and especially as last (29.5% vs. 23.4%) authors have increased in 2020 compared to our data covering 2010–2019. Similar findings were not evident between our and the earlier study covering 1994–2014. However, by examining bibliometric data of the Public Library of Science journals between 2010 and 2020, Giannos et al. have reported similar development despite variations by research fields. Collectively, the findings suggest that while gender representation has improved only recently, the progress is promising by being visible in multiple journals.

By comparing research keywords, men as the first and especially last authors tended to publish clinical trials and oncology-associated studies, which accumulated highest citation counts. While women have been shown to publish more qualitative studies and on primary healthcare, such a large difference in research focus has not been demonstrated before and likely accounts for some of the journal-specific gender disparities[22]. However, our data also indicated that publications first or last-authored by men tend to accumulate more citations implying that differences in research fields explain only limited variability.

In line with results of our study, articles with a woman researcher as first or last author have been shown to accumulate fewer citations[23]. While medical research was not included, a previous study using 1.5 million interdisciplinary papers in 1779–2011 has indicated that men as first authors tend to self-cite 56% themselves more commonly than women, and even more during the last decades[24]. According to a recent preprint, men were more commonly quoted in Nature science journalism, which could skew the recognition and future citation probability of publications by gender[25].

Between 2010 and 2019 articles the number of authors increased by nine reflecting a fundamental change in research towards larger collaboration and building consortia. This correlated with higher citation frequency per article. While the general

citation rate increased in the top medical journals, the rate was over 2.5 times faster in articles with more collaborators. We found that publications authored by men had more authors in concordance with a previous study[26]. In our study, both author gender and number of collaborators were independent but weak predictors of future citation rate.

Computational gender prediction could be a source of error as some names can be used by both women and men, especially in different countries. The name-based analysis may also misclassify authors with non-binary genders. We employed the commonly used Genderize.io database, which has been built on social media networks across 89 languages[27]. Despite Genderize.io performing well, its accuracy is suboptimal for Asian names[27]. However, recently reported gender-related bibliometric observations were in line with our findings indicating that the gender prediction based on authors' first names provided realistic results[15,17,21]. Inclusion of gender ethnicity and career stage as well as shared first or last authorships were unavailable but could be important factors to further study gender representation.

In summary, this computational audit indicated that gender disparity in medical research is country-specific, partly related to the distinct research focus and more evident at the senior researcher level. The study analyses can be easily replicated for any journals and any time period using data available at the Web of Science's database and codes published with this study. The analysis also highlighted that the gender gap is decreasing with country-dependent variability.

## Data availability
Source data for all figures and a 100-row example of the raw data from the Clarivate Web of Science is available at https://github.com/obruck/International-Research-Impact[28]. The full raw data can be downloaded from Clarivate Web of Science, with instructions provided in the Github repository.

## Code availability
Codes are available at https://github.com/obruck/International-Research-Impact[28].

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

## Acknowledgements
The author wishes to thank Susanna Lallukka-Brück for her patience, understanding, and insightful comments. The author is grateful to Olli Dufva and to the members of the Hematoscope Lab for discussion and comments. This study was supported by research grants from the Helsinki University Hospital. Certain data included herein are derived from Clarivate Web of Science. Open access funded by Helsinki University Library.

## Author contributions
Conception and design; Collection and assembly of data; Data analysis; Manuscript writing; Manuscript editing; Data interpretation; Final approval of manuscript: O.B.

## Competing interests
O.B. declares no competing non-financial interests but the following competing financial interests: consultancy fees from Novartis, Sanofi, and Amgen, outside the submitted work, and research grants from Pfizer and Gilead Sciences, outside the submitted work.
