## [Peer Review File · Communications Medicine]

Reviewers' comments:

Reviewer #1 (Remarks to the Author):

Dear Madams and Sirs,

Thank you for submitting the manuscript entitled:

The Gender Gap in Leading Medical Journals - a Computational Audi

In this study, the authors evaluated the bibliometric data on original research articles (n=10,558) published in 2010-2019 in five leading medical journals to audit publication and citation frequency by gender. As a result, they found out that there were fewer women as seniors (24.8%) than leading authors (34.5%, $p < 0.001$).

This manuscript has some significant writing issues, both grammatical and linguistic. The topic is not very new, and several studies deal with this critical issue, which the authors still need to quote. Unfortunately, the style of the text could be more professional and suitable for publication. The quotations/references are not by the journal's required reference style, especially when quoting from an URL. The figures need some resolution and more specific legends and descriptions of the images presented here. The tables provided must be formatted correctly; they are too crowded, the letters are not scientific and too confusing, and they should be better organized.

The study design needs to be organized, and there needs to be a clear description of the methods and how they have evaluated the journals. In the abstract, they are talking about Japan, but in the text, tables, and graphs, no numbers or sentences mention Japan. What are the authors presenting, then? The whole manuscript is abysmally organized. There are no structures whatsoever – please write the manuscript how it is supposed to be written for a journal. Also, how did the authors determine which names are female and male? Especially in Asian countries, not only in Japan but also in South Korea or China, the characters are complicated to interpret. Who did the analyses?

Some specific comments are as follows.

Abstract:

The abstract needs to be in concordance with the main text. At this point, accepting the manuscript for publication is impossible.

Introduction:

It needs to be longer. Newer studies should be mentioned. The authors should at least mention other studies dealing with this issue.

Material and Methods:

None existing

Results

This needs to be clarified. The tables need to be organized. The graphs are not explaining themselves.

Discussion:

Not existing.

Reviewer #2 (Remarks to the Author):

The article entitled "The Gender Gap in Leading Medical Journals - a Computational Audit" is a wide-ranging descriptive assessment of gender in publications in leading medical journals. This audit reveals that the disparity is largely country-specific as related to the distinct research foci of the authors. Although, the gender gap is decreasing with country-level variables. This disparity is more pronounced at the senior level, possibly related to access to resources.

Overall, this paper is well-written and offers a convincing suite of descriptive statistics to unpack the various ways in which the gender gap manifests in leading medical outlets. True to its title, the paper is an appropriate audit in this regard. There are two notes that I would like the paper to address in order to help develop the paper.

First, I would request that the paper's introduction is rewritten for clarity a bit. It's a bit unclear why a descriptive audit is the most appropriate approach, as well as just focusing on leading journals. I think the introduction could justify this and make the motivation a bit more strongly connected with the analyses and results. For instance, the paper's analyses appear to be focused on gender and nationality, and not really on their minoritized status. I think a bit more time dedicated to this would make the flow of the paper a bit crisper.

Second, is there an issue with just the five leading medical journals? Or, is there an issue with limiting the analyses to just elite journals, more generally? I certainly don't want to ask the authors to do more work, but I think some justification as to why these five journals and not including others would be helpful.

Reviewer #3 (Remarks to the Author):

Comments

I appreciate the effort that went into this study and please accept all comments as constructive with the intention of strengthening the paper. The authors of this study performed a bibliometric analysis to investigate the gender dynamics in the authorship of papers published in 5 high impact factor journals. The topic is very interesting and relevant and has value for the broader scientific community. The paper is well structured although some improvements can be made. The analysis is well described however, methodological limitations exist and should be discussed further in the discussion section.

Introduction

- The authors should consider mentioning that an inclusive academic environment ensures that potential is not lost. In other words, only by being inclusive we can reach our true academic and technological potential as a society.
- The authors should consider providing specific reasons for the selection of these five journals considering that impact factor does not necessarily echo the wider perception of researchers about the journals' content.

Methods

- The authors should mention the cut-off percentage used in the genderize.io database for the determination of author's gender. This is a crucial point considering that high cut offs can lead to the exclusion of specific names.
- Did the authors account for dual first author publications. If not, they have to mention it in the

methods and discussion sections of the manuscript.

Results

- Although interesting, I don't think that the paragraph about the trend in the number of authors is very relevant to the gender dynamics overall (lines 134-164). The issue regarding multiple authorships and the associated dilution effect is a general trend and the authors should consider transferring this section in the supplement.
- The authors mention "For first authors, 78/579 compared to 5/579 keywords resulted in higher citation count (adjusted $p < 0.05$) when employed by men vs. women, respectively." This covers around 15% (83/579) of the total keywords. My understanding is that for the remaining keywords the authors found non-significant differences. This should be clarified. Therefore, this result should be interpreted with caution.

Discussion

- The authors should consider discussing a recently published paper by Giannos et al on Female Dynamics in Authorship of Scientific Publications (doi: 10.3390/ejihpe13020018.). The fact that some of the results are in line with the current study provides additional credibility and gravitas.
- The authors should consider discussing the phenomenon of demographic inertia to explain the differences seen in last authors and citation count by gender. The shortage of women in senior positions within academia (demonstrated in the low number of female last authors) can be attributed to the fact that women scholars were fewer in the past. As a result, there may be a lag period of time before women reach more senior positions and attract more citations as senior authors.
- The authors should elaborate more on the limitations of genderize.io database. There is evidence that this tool has limited confidence in predicting gender names of Asian origin and this point should be mentioned.

I would like to express my sincere gratitude to the Reviewers for their thoughtful evaluation of the manuscript and sharing their insights to improve the scientific interest of the work and making it more accessible to a wide range of readers.

The main modifications are:

- Text editing to restructure the manuscript, extend the Introduction section and provide rationale for the study design and methods.
- Figures: Moving Fig. 2a-b to Supplementary Figures, joining Figures 2 and 3, and removing Supplementary Fig. 2.

Replies to Reviewer comments are highlighted in yellow. Here, I have marked removed words with ~~strikethroughs~~ and highlighted added words in red.

I have located the edits in the manuscript to the file named "Bruck_Manuscript_CommsMed_tracked_changes.docx". There, I have marked removed words with ~~strikethroughs~~ and highlighted added words in yellow.

Reviewers' comments:

Reviewer #1 (Remarks to the Author):

Dear Madams and Sirs,

Thank you for submitting the manuscript entitled:

The Gender Gap in Leading Medical Journals - a Computational Audit

In this study, the authors evaluated the bibliometric data on original research articles (n=10,558) published in 2010-2019 in five leading medical journals to audit publication and citation frequency by gender. As a result, they found out that there were fewer women as seniors (24.8%) than leading authors (34.5%, $p < 0.001$).

I would like to express my sincere appreciation for the time and effort the Reviewer dedicated to evaluating both manuscripts. I have taken each concern into careful consideration and made the necessary revisions to ensure that the manuscript meets the rigorous standards of scientific publication.

This manuscript has some significant writing issues, both grammatical and linguistic.

I appreciate the Reviewer's attention to the writing aspects of the paper and apologize for any grammatical or linguistic issues that may have affected the readability of the manuscript. The manuscript has been revised in detail and any writing errors have been corrected.

The topic is not very new, and several studies deal with this critical issue, which the authors still need to quote.

While the gender underrepresentation is not a new research field, few studies have examined high-impact medical journals or applied similar techniques (e.g. temporal analysis, keyword analysis, geographical analysis). However, I fully agree that recently many interesting articles on the topic have been published such as Sebo, P Fam. Pract (2020); Giannos, P. European Journal of Investigation in Health, Psychology and Education 13, 228–237 (2023); Shaw A Proceedings of the Royal Society B: Biological Sciences (2012); Hart K JAMA Internal Medicine (2019); Molwitz I. Scientometrics (2023), which have been cited in the revised version.

Unfortunately, the style of the text could be more professional and suitable for publication. The quotations/references are not by the journal's required reference style, especially when quoting from an URL. The figures need some resolution and more specific legends and descriptions of the images presented here. The tables provided must be formatted correctly; they are too crowded, the letters are not scientific and too confusing, and they should be better organized.

In the revised version, I have ensured that the language and style are refined to meet the standards expected for publication in this journal. Regarding the references, I have revised the manuscript to ensure that all references follow the prescribed format, including proper citation of URLs.

Necessary adjustments to figures and their legends have been made to enhance their clarity and support their interpretation. For example, Fig. 2a-b have been moved to the Supplementary Figures while Fig. 2 and 3 have been joined. I have aligned the formatting of the tables with the manuscript.

The study design needs to be organized, and there needs to be a clear description of the methods and how they have evaluated the journals.

In the revised manuscript, I have ensured that the study design is presented in a logical and organized manner, allowing readers to follow the research process easily.

To address the concern about the methods, I have extended especially procedures used to infer author gender based on names and restructured the Methods section.

In the abstract, they are talking about Japan, but in the text, tables, and graphs, no numbers or sentences mention Japan. What are the authors presenting, then?

I apologize for the confusion. Japan has been discussed as a representative example country in the Abstract section (L61), in the Results section (L210-212 and 215-216), in the Discussion section (L324-326), and Figures (Fig. 1g, Fig. 2a and Supplementary Fig. 1). Besides Japan, the manuscript presents results also on other countries such as Austria, South Korea, South Africa, India, Singapore, Germany, UK, China, Israel, USA, Denmark, Sweden, and France.

The whole manuscript is abysmally organized. There are no structures whatsoever – please write the manuscript how it is supposed to be written for a journal.

I apologize for the confusing manuscript structuring (e.g., Abstract; lack of a Plain Language Summary), arrangement (e.g., Main section; Results before Methods section) and selection of reference style (e.g., URL citation). To facilitate sending manuscripts for peer-review, many scientific journals, including Communications Medicine, do not require strict adherence to journal formatting requirements upon initial submission (<https://www.nature.com/commsmed/submit/submission-guidelines#submission-policies>). The formatting has now been fully revised to match the journal's guidelines.

Also, how did the authors determine which names are female and male? Especially in Asian countries, not only in Japan but also in South Korea or China, the characters are complicated to interpret. Who did the analyses?

I apologize for the confusion. I used the genderizeR R-programming library based on the Genderize.io database, where forenames have been annotated to be female or male using social networks data from 89 languages. The program has been widely used and with reliable results based on previous benchmarks studies (e.g. Santamaría L et Mihaljević H (2018)). In this study, there were in total 7,558 unique forenames of which the gender could be determined for 94.1%.

The reviewer raised a point, whether this would be more challenging for authors from Japan, South Korea or China where the characters are complicated to interpret. The names of all authors, regardless on their origin, are available in latin alphavet, so this

issue did not complicate the analysis. Although the tool gives slightly lower probabilities for Asian names, their reliability is mostly ~90-100% based on a study by Santamaría L.

The description in the Methods section L144-154 has been extended:

“Author name prediction

To determine the number of authors, we summed the frequency of the semicolon ";" delimiter between author names and then added 1. All author names were available in latin alphabet. The bibliometric data in the Web of Science does not specify possible shared first or last authorships. Therefore, our analysis interpreted first and second author names based on their sequential authorship order. For author names, only the initials of the forename were available for 538 first (5.1%), 529 second (5.0%), 780 second last (7.4%) and 572 last authors (5.4%). In total, we identified 7,558 unique forenames and defined the gender for 7,113 (94.1%) forenames with the genderizeR library based on the genderize.io database. To avoid bias related with names from certain countries, we applied a low (>50%) probability threshold.”

In addition, the performance of the computational method has been discussed in L381-390 referring to the prior benchmarking study and to similar results in other publications

“Computational gender prediction could be a source of error as some names can be used by both women and men, especially in different countries. The name-based analysis may also misclassify authors with non-binary genders. We employed the commonly used Genderize.io database, which has been built on social media networks across 89 languages²⁷. Despite Genderize.io performing well, its accuracy is suboptimal for Asian names²⁷. However, previously recently reported gender-related bibliometric observations were in line with our findings indicating that the gender prediction based on authors' first names provided realistic results^{15,17,21}. Inclusion of gender ethnicity and career stage as well as shared first or last authorships were unavailable but could be important factors to further study gender representation.”

Some specific comments are as follows.

Abstract:

The abstract needs to be in concordance with the main text. At this point, accepting the manuscript for publication is impossible.

I apologize for any inconsistencies that may have arisen between the abstract and the main text. I have revised and edited both the abstract and the main text, and can confirm their consistency and coherence.

Introduction:

It needs to be longer. Newer studies should be mentioned. The authors should at least mention other studies dealing with this issue.

I agree with the Reviewer that the Introduction should be longer. I have now fully revised the Introduction section L81-121 and added two entire new chapters citing essential previous works in the field.

“Gender equity in medical research refers to the equal representation, recognition, and valuation of all researchers regardless of their gender. Women and other underrepresented groups continue to face barriers in diverse research fields. The proportion of graduating female medical students is equal to that of men¹. Yet, publications are more frequently authored^{2,3}, peer-reviewed⁴, and editorially evaluated by men^{5,6}. Based on a report by the Association of American Medical Colleges, the proportion of women faculty had increased to 41% in 2019, while their proportion as department chairs remained at 18%⁷.”

Gender underrepresentation in scientific publishing has been documented earlier. Women tend to obtain university-level degrees more commonly than male in OECD countries (47% vs. 32%)⁸. However, female doctoral students submit and publish less than their male colleagues^{2,3}. The difference is largest in natural/biological sciences and engineering and has been hypothesized to result of unevenly distributed resources and support³. Gender disparities in publications have been shown to correlate with future academic rank signifying sustained impact on professional careers⁹, salary and job satisfaction¹⁰. Instead, pay transparency has been recently shown to significantly reduce inequity¹¹.

Abrogating the gender gap in all levels of academia are official priorities of the National Institutes of Health¹² and the European Commission¹³. Promoting gender equity can diversify science and be a critical step in achieving both academic and technological breakthroughs lead to more representative and balanced research. In addition, it can help to reduce bias and discrimination in academic careers by ensuring that all researchers are treated fairly and given equal opportunities to succeed.

~~We recently discovered previously undocumented Anglocentric bias in leading medical journals based on publication counts and their citation frequency¹⁴. We reasoned that gender underrepresentation could be studied with a similar approach and explore publication differences in research fields, scientific journals, and collaboration scope.~~

The representation of women in leading and senior publication authorship positions remains disproportionately low^{14,15}. Medical journals with high journals impact factors (JIF) have not been comprehensively examined, although these share demanding editorial and peer-review assessment, and priority for groundbreaking science. We recently discovered previously undocumented Anglocentric bias in leading medical journals based on publication counts and their citation frequency¹⁶. Here, we reasoned that gender underrepresentation could be objectively explored with a similar retrospective approach by examining research fields, geographical regions, scientific journals, and collaboration scope.”

Material and Methods:

None existing

If I understood correctly, the Reviewer did not raise any comments of the Methods in L124-177.

Results

This needs to be clarified. The tables need to be organized. The graphs are not explaining themselves.

I apologize for the confusion. The number of figures has been reduced and the figure legends have been edited. The formatting of the tables has been aligned.

Discussion:

Not existing.

If I understood correctly, the Reviewer did not raise any comments of the Discussion section in L303-398.

Collectively, I have addressed all comments raised by the Reviewer 1 with comprehensive revisions. I would like to thank the Reviewer's valuable feedback, which has helped to enhance the quality of the manuscript.

Reviewer #2 (Remarks to the Author):

The article entitled “The Gender Gap in Leading Medical Journals - a Computational Audit” is a wide-ranging descriptive assessment of gender in publications in leading medical journals. This audit reveals that the disparity is largely country-specific as related to the distinct research foci of the authors. Although, the gender gap is decreasing with country-level variables. This disparity is more pronounced at the senior level, possibly related to access to resources.

Overall, this paper is well-written and offers a convincing suite of descriptive statistics to unpack the various ways in which the gender gap manifests in leading medical outlets. True to its title, the paper is an appropriate audit in this regard. There are two notes that I would like the paper to address in order to help develop the paper.

Thank you sincerely for the thoughtful assessment and your kind and positive words. I greatly appreciate your time and efforts to review the manuscript, and I have carefully considered and addressed all comments.

First, I would request that the paper’s introduction is rewritten for clarity a bit. It’s a bit unclear why a descriptive audit is the most appropriate approach, as well as just focusing on leading journals. I think the introduction could justify this and make the motivation a bit more strongly connected with the analyses and results. For instance, the paper’s analyses appear to be focused on gender and nationality, and not really on their minoritized status. I think a bit more time dedicated to this would make the flow of the paper a bit crisper.

Second, is there an issue with just the five leading medical journals? Or, is there an issue with limiting the analyses to just elite journals, more generally? I certainly don’t want to ask the authors to do more work, but I think some justification as to why these five journals and not including others would be helpful.

I agree with the Reviewer. The Introduction section has been fully revised to cover previous findings, the social impact of gender underrepresentation and to provide clear justifications for selecting only five elite journals. These explanations are placed in the Introduction section in 91-121

“Gender underrepresentation in scientific publishing has been documented earlier. Women tend to obtain university-level degrees more commonly than male in OECD countries (47% vs. 32%)⁸. However, female doctoral students submit and publish less than their male colleagues^{2,3}. The difference is largest in natural/biological sciences and engineering and has been hypothesized to result of unevenly distributed resources and support³. Gender disparities in publications have been shown to correlate with future academic rank signifying sustained impact on professional careers⁹, salary and job satisfaction¹⁰. Instead, pay transparency has been recently shown to significantly reduce inequity¹¹.

Abrogating the gender gap in all levels of academia are official priorities of the National Institutes of Health¹² and the European Commission¹³. Promoting gender equity can diversify science and be a critical step in achieving both academic and technological breakthroughs lead to more representative and balanced research. In addition, it can

help to reduce bias and discrimination in academic careers by ensuring that all researchers are treated fairly and given equal opportunities to succeed.

~~We recently discovered previously undocumented Anglocentric bias in leading medical journals based on publication counts and their citation frequency¹⁶. We reasoned that gender underrepresentation could be studied with a similar approach and explore publication differences in research fields, scientific journals, and collaboration scope.~~

The representation of women in leading and senior publication authorship positions remains disproportionately low^{14,15}. Medical journals with high journals impact factors (JIF) have not been comprehensively examined, although these share demanding editorial and peer-review assessment, and priority for groundbreaking science. We recently discovered previously undocumented Anglocentric bias in leading medical journals based on publication counts and their citation frequency¹⁶. Here, we reasoned that gender underrepresentation could be objectively explored with a similar retrospective approach by examining research fields, geographical regions, scientific journals, and collaboration scope.”

In addition, the rationale for the included journals has also been edited in the Methods section L125-130

“For this study, we collected data from ~~five~~ medical journals that were ranked highest in the Journal Citation Reports 2022, ~~and~~ published primarily original articles ~~and within overlapping scientific scopes~~ (Fig. 1A). ~~These were~~ *The New England Journal of Medicine* (NEJM), *Nature Medicine* (NatMed), *Journal of the American Medical Association* (JAMA), *The BMJ*, and *The Lancet* ~~met these criteria.~~”

I would like to thank the Reviewer 2 for their comments, which have significantly contributed to the advancement of both manuscripts.

Reviewer #3 (Remarks to the Author):

Comments

I appreciate the effort that went into this study and please accept all comments as constructive with the intention of strengthening the paper. The authors of this study performed a bibliometric analysis to investigate the gender dynamics in the authorship of papers published in 5 high impact factor journals. The topic is very interesting and relevant and has value for the broader scientific community. The paper is well structured although some improvements can be made. The analysis is well described however, methodological limitations exist and should be discussed further in the discussion section.

I am very happy to share the excitement on the methodology and results of this manuscript the Reviewer #3, and would like to thank for the comments to further strengthen it.

Introduction

- The authors should consider mentioning that an inclusive academic environment ensures that potential is not lost. In other words, only by being inclusive we can reach our true academic and technological potential as a society.

I agree entirely with the comment. This has been covered in the Introduction section L91-104

“Gender underrepresentation in scientific publishing has been documented earlier. Women tend to obtain university-level degrees more commonly than male in OECD countries (47% vs. 32%)⁸. However, female doctoral students submit and publish less than their male colleagues^{2,3}. The difference is largest in natural/biological sciences and engineering and has been hypothesized to result of unevenly distributed resources and support³. Gender disparities in publications have been shown to correlate with future academic rank signifying sustained impact on professional careers⁹, salary and job satisfaction¹⁰. Instead, pay transparency has been recently shown to significantly reduce inequity¹¹.

Abrogating the gender gap in all levels of academia are official priorities of the National Institutes of Health¹² and the European Commission¹³. Promoting gender equity can diversify science and be a critical step in achieving both academic and technological breakthroughs lead to more representative and balanced research.”

- The authors should consider providing specific reasons for the selection of these five journals considering that impact factor does not necessarily echo the wider perception of researchers about the journals' content.

This is an essential perspective. The rationale has been clarified in the Introduction section L113-121

“The representation of women in leading and senior publication authorship positions remains disproportionately low^{14,15}. Medical journals with high journals impact factors (JIF) have not been comprehensively examined, although these share demanding editorial and peer-review assessment, and priority for groundbreaking science. We recently discovered previously undocumented Anglocentric bias in leading medical journals based on publication counts and their citation frequency¹⁶. Here, we reasoned

that gender underrepresentation could be objectively explored with a similar retrospective approach by examining research fields, geographical regions, scientific journals, and collaboration scope.”

Methods

- The authors should mention the cut-off percentage used in the genderize.io database for the determination of author's gender. This is a crucial point considering that high cut offs can lead to the exclusion of specific names.

I agree with the Reviewer. Published benchmark studies (e.g. Santamaría L et Mihaljević H (2018)) do not recommend any specific cut-offs, while these indicate that the tool gives lower probabilities for Asian names. To avoid bias related with arbitrary cut-offs, the analyses were conducted with a low cut-off strategy (>0.5). This has been clarified in L151-154

“In total, we identified 7,558 unique forenames and defined the gender for 7,113 (94.1%) forenames with the genderizeR library based on the genderize.io database. To avoid bias related with names common in distinct languages, we applied a low (>50%) probability threshold.”

- Did the authors account for dual first author publications. If not, they have to mention it in the methods and discussion sections of the manuscript.

The Reviewer raises an excellent point. The bibliometric data in Web of Science does not cover the actual first authorships implying that dual authorships cannot be extracted from the data. While I initially attempted to compensate for this by studying both the first and second author names, the dual authorship limitation has been reported in the Methods section L147-149

“The bibliometric data in the Web of Science does not specify possible shared first or last authorships. Therefore, our analysis interpreted first and second author names based on their sequential authorship order.”

And in the Discussion section L388-390

“Inclusion of gender ethnicity and career stage as well as shared first or last authorships were unavailable but could be important factors to further study gender representation.”

Results

- Although interesting, I don't think that the paragraph about the trend in the number of authors is very relevant to the gender dynamics overall (lines 134-164). The issue regarding multiple authorships and the associated dilution effect is a general trend and the authors should consider transferring this section in the supplement.

One of the main rational of this study was to understand the key factors related to gender underrepresentation in medical journals. While I agree with the Reviewer that the focus should not digress from the main storyline, this section gradually explains the essential association between author count, gender and geographical location, which have not been studied before in the presented context.

In the manuscript, I have removed Supplementary Figure 2, moved Figures 2a-b to Supplementary Fig. 2, and considerably shortened the section in L228-263.

“Collaboration scope varies by gender and is associated with publication and citation count

~~Next, we interrogated the significance of collaboration scope over gender-related differences in citation count. As the number of co-authors ($R=0.46$, $p<0.001$) correlated with citation count, we interrogated next the significance of collaboration scope over gender-related differences in citation count. Male first and last authors published in average with one author more than their female colleagues. On average, male first and last authors have published with one more author than their female colleagues (Table 1). To measure the independent impact of author count and gender on citations, we applied a linear regression model using these and their interaction term as covariates. In parallel, we examined the association of gender and citations in subgroups of categorized authorship count. Both approaches revealed that However, when applying a linear regression model using gender, author count and their interaction term as covariates, these were all observed to be **author count and gender are** independent but weak predictors of future citation count (Supplementary Table 1, **Supplementary Fig. 2**).~~

When examining the temporal evolution of authorship patterns, we observed a stable median increase of 9 additional authors per article during the 10-year follow-up (Supplementary Fig. 2a). Publications with more than 11 authors doubled ~~in that time~~ and these accumulated faster citations per article (2.99 citations/article/year vs. 1.19 citations/article/year; **Supplementary Fig. 2a-b**). The inclination was also 55.3% steeper for men as first author and 94.1% for men as last authors during the 10-year follow-up compared to corresponding female authors (Supplementary Fig. 2c). Instead, the slope between first and last authors did not differ when comparing separately women and men (Supplementary Fig. 2c).

To study national differences in collaboration scope, we compared the number of authors for publications by gender and by country (Fig. 2e). ~~While the number of authors did not differ for most countries, almost all variation in the number of authors was were related to the geographical origin of the research.~~ Publications from USA, UK, Denmark, and Sweden shared the least authors (Fig. 2e). On the opposite, publications from France, Germany, China, and South Korea were associated with more numerous authors/article (Fig. 2e). Together, the findings emphasize that the geographical origin of the research has a more pronounced association with collaboration scope than gender.”

- The authors mention “For first authors, 78/579 compared to 5/579 keywords resulted in higher citation count (adjusted $p<0.05$) when employed by men vs. women, respectively.” This covers around 15% (83/579) of the total keywords. My understanding is that for the remaining keywords the authors found non-significant differences. This should be clarified. Therefore, this result should be interpreted with caution.

The Result section in L287-292 has been revised

~~“Collectively, while the finding is relevant in distinct fields, no association between gender and citation count were observed for 85% of keywords when comparing first authors by gender and 93% of keywords when comparing last authors by gender. Collectively, the findings indicate that within the same fields of research, publications~~

authored by women accumulate fewer citations compared to publications authored by men.”

Discussion

- The authors should consider discussing a recently published paper by Giannos et al on Female Dynamics in Authorship of Scientific Publications (doi: 10.3390/ejihpe13020018.). The fact that some of the results are in line with the current study provides additional credibility and gravitas.

This study is indeed relevant by covering a similar time period and employing similar methods. Due to its overlap, this study has been visited in L349-353

“However, by examining bibliometric data of the Public Library of Science journals between 2010-2020, Giannos *et al* have reported similar development despite variations by research fields. Collectively, the findings suggest that while gender representation has improved only recently, the progress is promising by being visible in multiple journals.”

- The authors should consider discussing the phenomenon of demographic inertia to explain the differences seen in last authors and citation count by gender. The shortage of women in senior positions within academia (demonstrated in the low number of female last authors) can be attributed to the fact that women scholars were fewer in the past. As a result, there may be a lag period of time before women reach more senior positions and attract more citations as senior authors.

I agree with the Reviewer and this notion was alluded to in the initial Discussion section. Given that demographic inertia has been described previously, this section has been extended and a central publication also cited in L331-335

“The gender disparity has decreased more rapidly in last authorships possibly reflecting a response to the more pronounced gender underrepresentation compared to first authors or dynamics in generation transition (“demographic inertia”). This phenomenon could be due to a gradual increase of women in senior academic positions, while being rare in the past.”

- The authors should elaborate more on the limitations of genderize.io database. There is evidence that this tool has limited confidence in predicting gender names of Asian origin and this point should be mentioned.

Excellent point. This has been discussed in L381-386 and a benchmark study by Santamaría L et Mihaljević H (2018) has been referred to

“Computational gender prediction could be a source of error as some names can be used by both women and men, especially in different countries. The name-based analysis may also misclassify authors with non-binary genders. We employed the commonly used Genderize.io database, which has been built on social media networks across 89 languages²⁶. Despite Genderize.io performing well, its accuracy is suboptimal for Asian names²⁶.”

To conclude, I would like to sincerely thank the Reviewer for their innovative and thoughtful comments, which substantially improved the manuscript. All the comments have been addressed and the manuscript edited accordingly.

REVIEWERS' COMMENTS:

Reviewer #2 (Remarks to the Author):

I appreciate the author's effort in addressing the comments and feedback from me and the other reviewers. This will make a fine contribution to the gender gap in medical research. I have no further comments. Best of luck!

I think the author adequately addressed the critiques in both papers from Reviewer 1. (I'll add the caveat that I can't say whether Reviewer 1 would actually be satisfied with the edits, but from my point of view the author did a decent job in both papers.)

Reviewer #3 (Remarks to the Author):

The amendments are satisfactory. The manuscript reads well and the structure has been significantly improved. My comments have been addressed sufficiently.